# The Function of N-Myc Downstream-Regulated Gene 2 (NDRG2) as a Negative Regulator in Tumor Cell Metastasis

**DOI:** 10.3390/ijms23169365

**Published:** 2022-08-19

**Authors:** Ki Won Lee, Seyeon Lim, Kwang Dong Kim

**Affiliations:** 1Plant Molecular Biology and Biotechnology Research Center (PMBBRC), Gyeongsang National University, Jinju 52828, Korea; 2Division of Applied Life Science (BK21), Gyeongsang National University, Jinju 52828, Korea; 3Division of Life Sciences, Gyeongsang National University, Jinju 52828, Korea

**Keywords:** NDRG2, metastasis, epithelial–mesenchymal transition, tumor-suppressor

## Abstract

N-myc downstream-regulated gene 2 (NDRG2) is a tumor-suppressor gene that suppresses tumorigenesis and metastasis of tumors and increases sensitivity to anti-cancer drugs. In this review, we summarize information on the clinicopathological characteristics of tumor patients according to NDRG2 expression in various tumor tissues and provide information on the metastasis inhibition-related cell signaling modulation by NDRG2. Loss of NDRG2 expression is a prognostic factor that correlates with TNM grade and tumor metastasis and has an inverse relationship with patient survival in various tumor patients. NDRG2 inhibits cell signaling, such as AKT-, NF-κB-, STAT3-, and TGF-β-mediated signaling, to induce tumor metastasis, and induces activation of GSK-3β which has anti-tumor effects. Although NDRG2 operates as an adaptor protein to mediate the interaction between kinases and phosphatases, which is essential in regulating cell signaling related to tumor metastasis, the molecular mechanism of NDRG2 as an adapter protein does not seem to be fully elucidated. This review aims to assist the research design regarding NDRG2 function as an adaptor protein and suggests NDRG2 as a molecular target to inhibit tumor metastasis and improve the prognosis in tumor patients.

## 1. Introduction

N-myc downstream-regulated gene 2 (NDRG2) belongs to the NDRG family along with NDRG1, -3, and -4, forms a homologous cluster in several species, and is characterized by an NDR protein domain consisting of an esterase-/lipase-/thioesterase-active-site serine and an α/β-hydrolase fold of approximately 220 amino acids. However, NDRG proteins have no enzymatic activity [1,2,3]. NDRG2 has been proposed as a functional gene in response to cellular stresses in cellular metabolic processes [4,5,6], hypoxia [7], and lipotoxicity [8]. Additionally, the functions of NDRG2 as a tumor suppressor contribute to tumor growth inhibition and anti-metastasis in various tumors. Tumor metastasis is known to be the main cause of death in cancer patients [9]. Metastasis is the process of moving from the primary tumor to other organs due to the limited supply of oxygen and nutrients and the epithelial–mesenchymal transition (EMT) step is a key step for metastasis [10,11,12,13]. In this review, we aim to provide useful information for tumor control studies by summarizing clinicopathological properties in various tumors and presenting the function of NDRG2 as a modulator of cell signaling for EMT.

## 2. Anti-Tumor Function of NDRG2 Based on Clinicopathological Characteristics of Various Tumor Patients

The expression of NDRG2 seems more reasonable to be suggested as a prognostic marker related to the reduction in pathological symptoms of the tumor rather than a diagnostic marker related to tumor development. NDRG2 expression is regulated at the transcriptional or the post-transcriptional step. Hypermethylation of the NDRG2 promoter region is a representative mechanism of NDRG2 transcriptional repression in various tumors [14,15,16,17,18,19]. Another mechanism regulating NDRG2 expression is mediated by microRNAs (miRNAs); miRNA-181c [20], miR-454 [21], miR-483 [22], miR-650 [23], and et al. NDRG2 expression level is positively correlated with tumor differentiation and overall survivals, but negatively correlated with tumor invasion, tumor recurrence, lymph node metastasis, and TNM staging in various tumor patients (Table 1).

## 3. Anti-Metastatic Role of NDRG2

### 3.1. Epithelial–Mesenchymal Transition (EMT) and NDRG2

Cancers have the potential to metastasize and most cancer patients die from uncontrolled metastasis [46,47]. Therefore, establishing a strategy to control or inhibit tumor metastasis is very important for increasing the survival rate and treatment of tumor patients. Metastasis is a multi-step process initiated by the escape of tumor cells from primary tumor tissue and ending with the colonization of secondary tumors at distant tissues from the primary site. The steps of metastasis can be subdivided into the escape of cancer cells from the primary tumor, intravascular invasion, survival, extravasation, and colonization [48,49,50,51]. EMT is a key mechanism for promoting cancer metastasis, whereby cell types with a mesenchymal phenotype generally become invasive [52]. EMT is regulated by activation of the SNAI, ZEB, and TWIST transcription factor network (EMT-TF) families, which directly repress epithelial marker genes involved in cell adhesion, polarity, and cytoskeletal organization [53]. Various cellular signaling pathways, such as TGF-β signaling and AKT, NF-κB, and STAT3 activation, can induce the expression of EMT-TFs and metastatic effector proteins [54,55,56,57,58,59]. NDRG2 expression correlates inversely with TNM grade and metastasis in tumor patients (Table 1) and inhibits invasion and metastasis of tumor cells by modulating metastatic cell signaling pathways.

### 3.2. AKT and NDRG2

Bone is a common metastasis site in various cancer patients [60,61,62]. The phosphoinositide 3-kinase (PI3K)/AKT signaling pathway has been implicated in bone metastasis in several tumors. pAKT levels are elevated in 81.8% of primary HER2-positive breast cancers with bone metastasis, suggesting that the HER2/CXCR4/AKT pathway plays a role in bone metastasis [57]. The function of the CXCL12/CXCR4 axis in promoting metastasis is mediated in part by AKT activation [63,64]. In addition, bone metastasis is more frequently induced by breast cancer cells with hypoxia-inducing factor 1α (HIF-1α), which responds to hypoxia and is stabilized by AKT, resulting in CXCR4 upregulation [65]. Further, overexpression of AKT1 in prostate cancers upregulates CXCR4 expression and loss of PTEN in the cells enhances AKT-mediated expression of CXCL12 and CXCR4 [66]. Low NDRG2 expression is associated with pAKT and XIAP upregulation, while overexpression of NDRG2 suppresses AKT/XIAP signaling pathway and EMT in esophageal cancer cells [34]. NDRG2 is considered a PTEN-binding protein recruiting protein phosphatase 2A (PP2A) to PTEN, which inhibits PI3K-AKT pathway via PTEN activation by PP2A in adult T-cell leukemia-lymphoma (ATLL) (Figure 1) [67]. Therefore, it is possible NDRG2 inhibits metastasis mediated by AKT activation pathway by acting as a bridge between PP2A and PTEN to inhibit PI3K activation.

### 3.3. Nuclear Factor-κB (NF-κB) and NDRG2

NF-κB is a key transcription factor contributing to pathogenesis of inflammation and cancer [68]. NF-κB forms a complex with IκB in the cytoplasm, and when IκBα is ubiquitinated and subsequently degraded by the 26S proteasome, NF-κB is released and migrates to the nucleus to act as a transcription factor. Specifically, NF-κB activation is a potential mechanism explaining the relationship between inflammation and tumors [69]. Metastatic breast cancer cells overexpress the chemokine receptor CXCR4, and metastatic tissues express a large amount of CXCL12, a ligand of CXCR4 [48]. Further, NF-κB activation induces expression of matrix metalloproteinases, urokinase-type plasminogen activators, and cytokines in metastatic breast cancer cell lines [58]. NF-κB p65 (RelA) is constitutively activated in gastric cancer, which correlated with cancer invasion-associated clinicopathological features, lymphoid invasion, depth of invasion, and peritoneal metastasis [70]. Activation of NF-κB induces osteolytic bone metastasis by inducing *CSF2* (GM-CSF) expression [71]. Loss of NDRG2 expression is associated with tumor metastasis by inducing EMT via NF-κB activation [72]. NDRG2 attenuated NF-κB activation in PMA-treated human breast cancer cell (MDA-MB-231), which suppresses migration and invasion by inhibiting COX-2 expression [73]. Furthermore, NDRG2 suppresses tumor invasion by attenuating matrix metalloproteinase 9 (MMP9) expression via NF-κB inhibition [74].

Although NDRG2 expression likely inhibits NF-κB-mediated tumor metastasis, how NDRG2 inhibits NF-κB activity remains unclear. Recently, Ichikawa et al. proposed NDRG2 inhibits NF-κB in canonical and non-canonical pathways [75]. In the canonical pathway, the PI3K-AKT signaling pathway enhances the activation of canonical NF-κB [76,77] and NDRG2-mediated PTEN activation may inhibit canonical NF-κB by attenuating the PI3K-AKT activation pathway. In the non-canonical NF-κB pathway, activated NF-κB-inducing kinase (NIK) phosphorylates IKKα, which leads to phosphorylation of p100 to partially degrade p52 formation. RelB-bound p52 translocates into the nucleus to induce target genes (Figure 2) [78,79,80]. NDRG2 inhibits non-canonical NF-κB by interacting with NIK, followed by dephosphorylation of NIK via the NDRG2–PP2A complex.

### 3.4. Signal Transducer and Activator of Transcription 3 (STAT3) and NDRG2

Activation of STAT proteins is typically induced by Janus-Kinase (JAK) family proteins which are receptor-associated tyrosine kinases [81]. Activation of STAT3 occurs in a variety of tumors and aberrant STAT3 activation is associated with tumorigenesis and invasion promotion [82,83,84,85]. Metastatic spread of tumor cells requires cell motility, extracellular matrix infiltration, and angiogenesis. STAT3 activation contributes to enhancement of integrin 6 expression that promotes tumor cell motility, acquisition of invasive traits such as MUC1, Bcl6, cathepsins, and UPA, and increases expression of MMPs [86,87,88]. Furthermore, expression of E-cadherin, a representative anti-metastatic phenotype, is inhibited by STAT3 activation. STAT3 regulates ZEB1 expression, which can participate in STAT3-induced cell invasion and E-cadherin downregulation in colorectal cancer [89]. TWIST, an EMT-TF, is transcriptionally induced by STAT3 activation and plays a role as a transcriptional repressor of E-cadherin [59,90]. STAT3 also contributes to angiogenesis through the action of a transcription factor inducing vascular endothelial growth factor (VEGF) expression [91,92]. In unstimulated cells, STAT3 is regulated by negative regulators, protein inhibitors of activated STATs (PIAS), members of the cytokine signaling inhibitor (SOCS) family, protein tyrosine phosphatases (SHP1, SHP2, PTPN1, PTPN2 PTPRD, and PTPRT), and ubiquitin enzymes, to remain inactive in the cytoplasm [93,94]. *SOCS* expression can be induced by STAT3 activation, after which SOCS becomes a STAT3 regulator [95]. Overexpression of NDRG2 can inhibit STAT3 activation by inducing SOCS1 [96] and inhibition of STAT3 by NDRG2 suppresses EMT by reducing SNAIL expression (Figure 3) [97]. The transcription factor MYB activates STAT3 and AKT pathways by inhibiting NDRG2 expression via miR-130a expression in salivary adenoid cystic carcinoma [98]. Although NDRG2 regulates on STAT3 and SOCS1 induction in NDRG2-expressed cells, which is considered a mechanism for NDRG2-mediated STAT3 inhibition, the molecular mechanism is still unclear. Further analysis into the molecular mechanisms is needed, such as (1) how does NDRG2 induce SOCS1 expression, and (2) can NDRG2 play a role as a bridge to allow JAKs or STAT3 to interact with protein tyrosine phosphatases?

### 3.5. TGF-β and NDRG2

Although TGF-β plays a tumor-suppressor function, in the later stages of cancer progression, cancer cells remain responsive to TGF-β but become resistant to the cell proliferation inhibitory effects by TGF-β. Here, TGF-β acts directly on the cancer cells, leading to EMT through induction of E-cadherin transcriptional repressors, such as SNAIL, ZEB, and TWIST [99,100]. TGF-β binds to the receptor and induces formation of a complex with Smad2, Smad3, and Smad4, which regulates the expression of various target genes [101]. Further, TGF-β-induced EMT is closely related to proteolytic activity of MMPs [102]. Among the MMP family, MMP2 plays an important role in basement membrane remodeling by degrading collagen I, IV, V, plasminogen, and laminin-5 [103,104,105]. Further, NDRG2 can antagonize TGF-β-mediated liver cancer cell invasion by downregulating expression of MMP2 and laminin 332 pathway components [41]. Meanwhile, integrin complexes, integrin αν and integrin β6, induce conversion from latent TGF-β into its active form [106]. In a mouse breast cancer cell model, overexpression of NDRG2 disrupts TGF-β-mediated gene expression and inhibits conversion from latent TGF-β by reducing of integrin β6 expression, which inhibits tumor cell invasion (Figure 4) [28]. Despite growing cell biological and clinical evidence on the relationship between NDRG2 expression and inhibition of TGF-β-mediated tumor metastasis, its molecular mechanism is yet to be identified.

### 3.6. GSK-3β and NDRG2

Overexpression of constitutively active GSK-3β mutants increases chemosensitivity, cell cycle arrest, and decreases tumorigenicity [107,108]. GSK-3 can inhibit the Wnt/β-catenin pathway by phosphorylating β-catenin, leading to ubiquitin/proteasome-dependent degradation of β-catenin that serves as a transactivator for TCF/LEFs. Therefore, GSK-3 acts as a tumor suppressor by inhibiting β-catenin-mediated tumorigenic characteristics including EMT [109,110]. AKT induces inactivation of GSK-3β, which in turn inhibits SNAIL phosphorylation, leading to SNAIL translocation into the nucleus and its stabilization leading to EMT [111,112]. Phosphoinositide 3-kinase (PI3K)-mediated AKT activation signaling is inhibited by phosphatase and tensin homolog (PTEN) which dephosphorylates the lipid second messenger, phosphatidylinositol 3,4,5-trisphosphate [113]. The GSK-3β mechanism activation resulting in EMT inhibition in NDRG2-overexpressed cells can be considered as follows: (1) NDRG2 as a PTEN-binding protein recruits PP2A to PTEN, which activates PTEN and inhibits PI3K-AKT signaling [67]; (2) NDRG2 forms a complex with PP2A and GSK-3β, then induces the activation of GSK-3β by removing inhibitory phosphorylation of GSK-3 β by PP2A (Figure 5) [114].

## 4. Conclusions and Outlook

Although expression of NDRG2 is not abolished in all carcinomas and tumor tissues, low expression of NDRG2 is closely related to poor prognosis, such as increased metastasis, higher TNM stage, and reduced survival. specifically, metastasis of tumor cells can worsen patient prognosis. Therefore, tumor metastasis inhibition may be a therapeutic strategy to improve prognosis and improve patient survival rate. Studies on NDRG2-mediated inhibition of tumor metastasis can provide useful scientific information on development of tumor therapeutics.

Although NDRG2 is a protein without enzymatic activity, it is involved in many cellular signaling pathways. We propose the role of NDRG2 bridging in mediating interactions between different proteins is crucial in modulating NDRG2-mediated cellular signaling. As an adapter protein, NDRG2 is already known to mediate AKT inhibition, NF-κB inhibition, and GSK3 activation by mediating PP2A/PTEN, PP2A/NIK and PP2A/GSK3 interactions, respectively. However, to clarify the molecular mechanism of NDRG2 related to the anti-tumor phenotype, further analysis of the interactions between kinase and NDRG2 located upstream of cell signaling, as well as interaction between NDRG2 and various phosphatases, such as serine/threonine phosphatases and tyrosine phosphatases, is needed. In particular, although a direct inhibitory mechanism for receptor tyrosine kinases (RTK) by NDRG2 has not been reported so far, we suggest research on the NDRG2–tyrosine phosphatase complex is necessary as a regulatory mechanism for RTK.

## Figures and Tables

**Figure 1 ijms-23-09365-f001:**
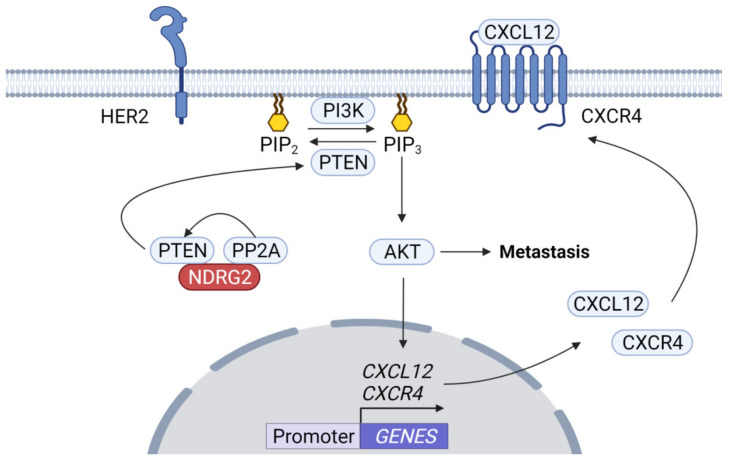
**AKT inhibition by the NDRG2–PP2A complex.** HER2 activates PI3K/AKT signaling. PTEN is a dual protein/lipid phosphatase using phosphatidylinositol, 3,4,5 triphosphate (PIP3), a product of PI3K, as a substrate. An increase in PIP3 recruits AKT to the membrane, which is activated by other PIP3-dependent kinases. NDRG2 acts as a bridge mediating the interaction between PP2A and PTEN, thus the NDRG2–PP2A complex recruits PTEN and triggers its activation. PTEN can inhibit AKT activity by inducing dephosphorylation of PIP3, thereby inhibiting AKT-dependent tumor cell metastasis.

**Figure 2 ijms-23-09365-f002:**
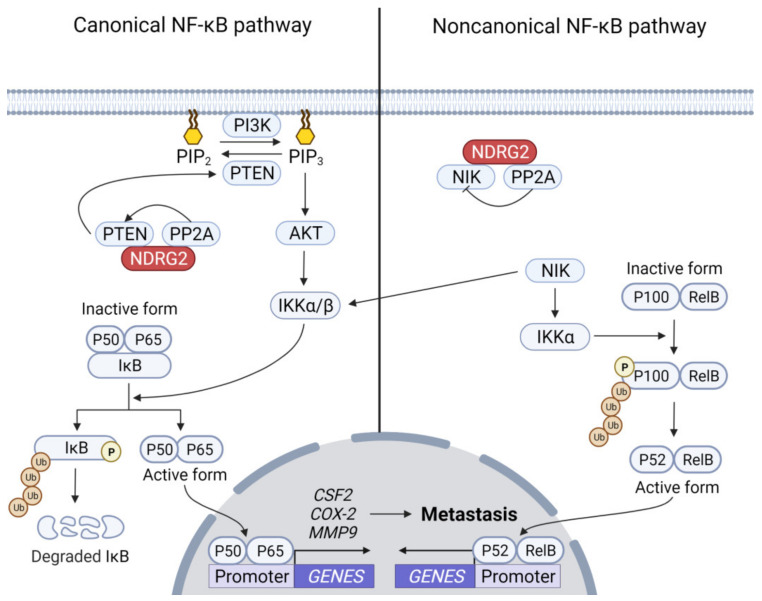
**Inhibition of NF-****κB-dependent tumor cell metastasis by NDRG2.** The PI3K/AKT/IKKα/NF-κB pathway contributes to tumor cell metastasis by inducing related genes, such as *CSF2*, *COX-2*, and *MMP9*. IKKα is also a substrate of NIK. NDRG2 inhibits the canonical NF-κB pathway via AKT inhibition to inhibit IKKα activation. Activated NIK induces phosphorylation of IKKα, which phosphorylates p100 to process p100 to p52 in a proteasome-dependent manner. The NDRG2–PP2A complex dephosphorylates NIK to inhibit the non-canonical NF-κB pathway.

**Figure 3 ijms-23-09365-f003:**
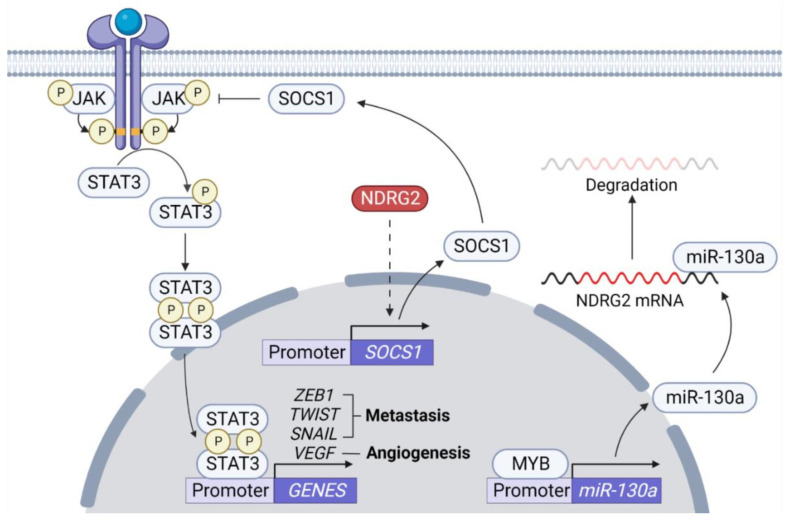
**Inhibition of STAT3 activation pathway by NDRG2.** JAK is activated upon cytokine or growth factors, resulting in dimerization through phosphorylation and nuclear translocation of STAT3 to induce transcription of metastasis-related genes, such as *ZEB1*, *TWIST*, *SNAIL*, and *VEGF*. Cytokine signaling inhibitory factor (SOCS) protein is a representative negative regulator on JAK. NDRG2 induces *SOCS1* expression through an unknown mechanism (dash line) and SOCS1 inhibits STAT3-mediated tumor cell metastasis by inhibiting JAK activation.

**Figure 4 ijms-23-09365-f004:**
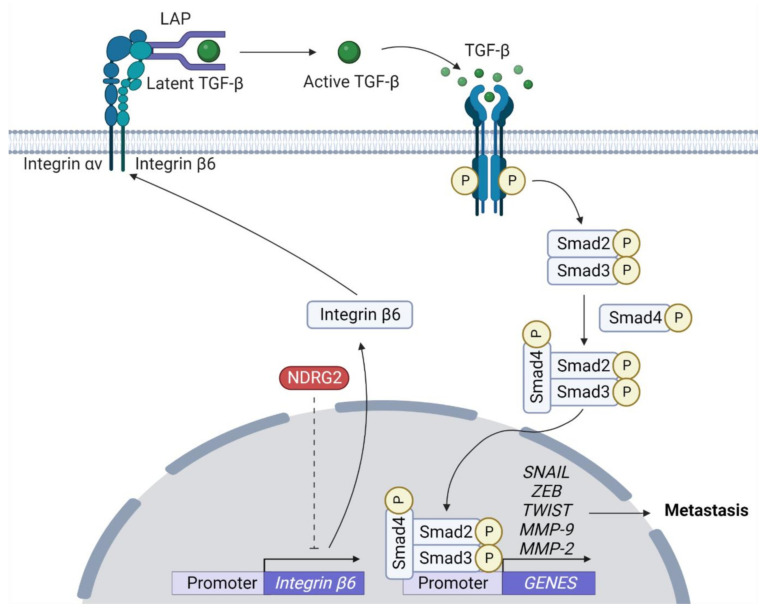
**NDRG2-mediated inhibition of TGF-β signaling**. αvβ6 activates TGF-β by binding to tripeptide Arg-Gly-Asp (RGD) motif of the latency-associated peptide (LAP). When integrins bind to the RGD motif of LAP, their binding to the actin cytoskeleton triggers a conformational change that releases active TGF-β. TGF-β via SMAD signaling can induce expression of EMT-TFs, such as SNAIL, TWIST, and ZEB, and MMP2. NDRG2 inhibits the conversion of latent TGF-β to active TGF-β by inhibiting expression of integrin β6 through an unknown mechanism (dash line).

**Figure 5 ijms-23-09365-f005:**
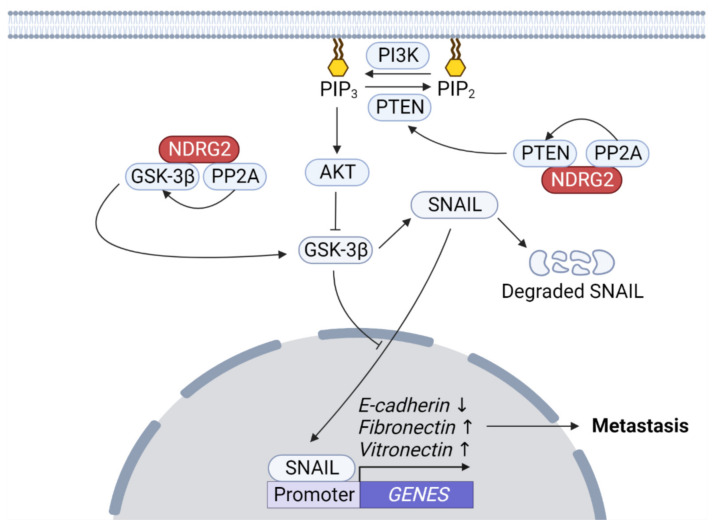
**NDRG2–PP2A-mediated GSK-3β activation.** AKT induces inhibitory phosphorylation of glycogen synthase kinase-3 beta (GSK-3β). The NDRG2–PP2A complex recruits PTEN, which dephosphorylates PIP3 to inhibit AKT activation. Further, the NDRG2–PP2A complex releases the inhibitory phosphate of GSK-3β to activate it. Active GSK-3β phosphorylates SNAIL to induce β-Trcp-mediated ubiquitination and degradation, which controls its subcellular localization. GSK-3β activation downregulates EMT phenotypes, such as E-cadherin upregulation and fibronectin and vitronectin downregulation.

**Table 1 ijms-23-09365-t001:** Clinicopathological role of NDRG2 in tumors.

Tumor	Reported Findings	Refs.
Astrocytoma	NDRG2 expression was negatively correlated with pathological grading but positively with the life span of astrocytoma patients.	[24,25]
Bladder cancer	The NDRG2 level was negatively correlated with TNM grade, pathologic stage, and increased c-myc level.	[26]
Breast cancer	Basal-like tumors more frequently maintained abundant NDRG2 expression consistent with lower CpG-hypermethylation, and NDRG2 expression was associated with a relatively good prognosis for patients.	[27]
	NDRG2 expression is negatively correlated with favorable recurrence-free survival in breast cancer patients.	[28]
	Patients with high NDRG2 expression have higher disease-free survival and overall survival than patients with low NDRG expression, and NDRG2 expression was negatively correlated with GLUT1 expression.	[4]
Colorectal cancer	NDRG2 was downregulated in colorectal cancer compared to benign colorectal tissues. The NDRG2 expression is regulated by promoter methylation and miR-650.	[23]
	Methylation of NDRG2 promoter is methylated in 89% of human CRC tissues compared to the adjacent normal colonic mucosa. Higher levels of the promoter methylation are more prevalent in proximal CRC and advanced T stage.	[29]
	Low NDRG2 expression presents a significantly poorer 3-year overall survival rate in patients with stage IV, liver metastasis, and those receiving liver resection.	[30]
	NDRG2 expression levels were significantly decreased in both adenomas (*p* < 0.001) and colorectal cancer, and NDRG2 levels tend to decrease with increasing stages of Dukes.	[31]
Esophageal cancer	Low expression of NDRG2 in ESCC patients is inversely associated with clinical stage, TNM classification, histologic differentiation, and patient vital status. During the 5-year follow-up period, the survival period of ESCC patients with high NDRG2 expression was longer than that of patients with low expression.	[32,33,34]
Gallbladder cancer	Loss of NDRG2 expression is an independent predictor of decreased survival, is associated with severe progressive T stage, lymphatic invasion, and induces expression of MMP-19, which regulates *Slug* expression at the transcriptional level.	[35]
	Downregulation of NDRG2 and upregulation of CD24 show more frequent lymph node metastasis and lymphovascular invasion in gallbladder cancer patients.	[36]
Gastric cancer	NDRG2 expression is regulated by DNA methylation and the downregulation of NRG2 expression is related to lymph node metastasis, tumor invasion, Borrmann classification and TNM stage in gastric cancer patients.	[37]
	Loss of NDRG2 expression is an important and independent prognostic indicator by multivariate analysis, which affects survival rate in gastric cancer patients.	[38]
	NDRG2 promoter methylation is triggered by *H. pylori* infection and downregulation of NDRG2 expression is associated with worse prognosis, which is an independent prognostic factor for the disease-free survival of gastric tumor patients.	[15]
Glioma	A Kaplan–Meier analysis of glioma patients showed improved survival time in the patients with mRNA and protein expression, and NDRG2 downregulation may affect glioma tumor progression to be higher malignancy.	[19]
	NDRG2 gene transcript is expressed in normal bran and low-grade glioma but is very low in glioblastoma, the most severe form of brain glioma.	[39]
	NDRG2 expression is inversely associated with lower survival rates and is an independent prognostic factor for overall survival in glioma patients.	[40]
Liver cancer	Reduction of NDRG2 is associated with its promoter hypermethylation and significantly correlated with tumor-node-metastasis stage, differentiation grade, portal vein thrombi, infiltrative growth pattern, nodal/distant metastasis, recurrent tumor, and shorter survival rates in liver cancer patients.	[41]
	Low expression of NDRG2 is strongly associated with tumor metastasis and high expression of NDRG2 is correlated with higher survival rates of hepatoblastoma patients.	[17]
	Downregulation of NDRG2 is strongly correlated with CD24 overexpression and is observed in hepatocarcinoma patients with elevated AFP serum level, late TNM stage, poor differentiation grade, tumor invasion, and recurrence.	[42]
	NDRG2 inhibits cholangiocarcinoma cell proliferation, chemoresistance, and metastasis, NDRG2 is a target of miR-181c which can be activated by leukemia inhibitory factor.	[20]
Lung cancer	Lung adenocarcinoma patients showing NDRG2-low/CD24-high expression have the lowest survival rate, and NDRG2-high/CD24-low and NDRG2-low/CD24-high expression patterns are independent prognostic indicators of lung adenocarcinoma.	[43]
Renal cancer	Loss of NDRG2 expression is an independent poor prognostic factor for renal cell carcinoma patients.	[44]
	NDRG2 expression is downregulated in clear cell renal cell carcinoma, which is negatively associated with aggressive tumor behaviors such as TNM stage, Fuhrman grade, tumor invasion, shorter patient survival, and tumor recurrence. In addition, NDRG2 can inhibit cancer cell invasion by regulating MMP-9 expression and activity.	[45]

## Data Availability

Not applicable.

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
