# Peer review of "The Function of N-Myc Downstream-Regulated Gene 2 (NDRG2) as a Negative Regulator in Tumor Cell Metastasis"

_ijms, 2022, doi:10.3390/ijms23169365_

Round 1

Reviewer 1 Report

I think is a nice review, but I would like the authors explain better the beahaviour of the gen in Gliomas

Author Response

I am appreciates for improving this manuscript with your comments.

Reviewer’s comment 1.: I think is a nice review, but I would like the authors explain better the behavior of the gen in Gliomas.

: Following advice, we added two references on the association of NDRG2 expression with clinical characteristics in glioma patients in Table 1.

Reviewer 2 Report

This is an interesting review aiming to assess the association between NDRG2 as an inhibitor of tumor metastasis. The review is quite comprehensive, and authors have also referred to latest papers to cite their statements. Nonetheless, I have several remarks:

1.      The introduction section needs to be rewritten as it doesn’t provide the overall concept that the authors want to address in their paper.

2.      Reframe the sentence “NDRG2 expression level is positively correlated with tumor differentiation and overall survivals of disease-free survivals”.

3.      Authors mentioned that “NDRG2 expression correlates inversely with TNM grade and metastasis in tumor patients (Table1) and inhibits invasion and metastasis of tumor cells through modulation of metastatic cell signalings” that suggests the role of NDRG2 in regulation of EMT. It would be great if authors could cite some literature relevant to this and further discuss the role of NDRG2 in EMT.

4.      Are there any known mechanisms for loss of function of NDRG2 in various human cancers. If there are any? It’s critical to mention them in the form of a table.

5.      There are few typo or spelling errors in the manuscript that needs to be addressed (e.g., ibhibition, etc.)

6.      The authors must avoid repetition of information; this will make the article more comprehensive, and the language of the paper should be checked by a native English speaker to make the paper fluent so that the readers don’t loose the interest while going through it.

Author Response

I am appreciates for improving this manuscript with your comments.

Reviewer’s comment 1.: The introduction section needs to be rewritten as it doesn’t provide the overall concept that the authors want to address in their paper.

: Following advice, The second half of the introduction has been rewritten. The following sentences have been added; “Tumor metastasis is known to be the main cause of death in cancer patients [9]. Metastasis is the process of moving from the primary tumor to other organs due to the limited supply of oxygen and nutrients and the epithelial-mesenchymal transition (EMT) step is a key step for metastasis [10-13]. In this review, we aim to provide useful information for tumor control studies by summarizing clinicopathological properties in various tumors and presenting the function of NDRG2 as a modulator of cell signaling for EMT.”

Reviewer’s comment 2.   Reframe the sentence “NDRG2 expression level is positively correlated with tumor differentiation and overall survivals of disease-free survivals”.

: Following advice, “of disease-free survivals” is deleted in the sentence.

Reviewer’s comment 3. Authors mentioned that “NDRG2 expression correlates inversely with TNM grade and metastasis in tumor patients (Table1) and inhibits invasion and metastasis of tumor cells through modulation of metastatic cell signalings” that suggests the role of NDRG2 in regulation of EMT. It would be great if authors could cite some literature relevant to this and further discuss the role of NDRG2 in EMT.

: The review's advice is a very important point, and I think it's the main topic covered in this review article. The content of this text (from 3.2 to 3.6) is about the NDRG2-mediated EMT regulation pointed out by the reviewer, and I think that there are sufficient references.

.Reviewer’s comment 4.  Are there any known mechanisms for loss of function of NDRG2 in various human cancers. If there are any? It’s critical to mention them in the form of a table.

: The reviewers’ comment is very important. It is known that methylation of NDRG2 promoter region in various tumor patients (breast cancer, colorectal cancer, stomach cancer, and liver cancer) is very important for regulating NDRG2 expression, and it is summarized in Table 1.

Reviewer’s comment 5 & 6.  There are few typo or spelling errors in the manuscript that needs to be addressed (e.g., ibhibition, etc.). The authors must avoid repetition of information; this will make the article more comprehensive, and the language of the paper should be checked by a native English speaker to make the paper fluent so that the readers don’t loose the interest while going through it.

: Following advice, this manuscript was finalized through proofreading services by native speakers (Crimson Interactive Inc.; www. Enago.co.kr).